# HSPCs display within-family homogeneity in differentiation and proliferation despite population heterogeneity

Tamar Tak[1†], Giulio Prevedello[2,3†], Gaël Simon[1], Noémie Paillon[1], Camélia Benlabiod[4,5,6], Caroline Marty[3,4,5], Isabelle Plo[3,4,5], Ken R Duffy[7], Leïla Perié[1]*

[1]Institut Curie, Université PSL, Sorbonne Université, CNRS UMR168, Laboratoire Physico Chimie Curie, Paris, France; [2]Institut Curie, PSL Research University, CNRS, Orsay, France; [3]Université Paris-Saclay, Saclay, France; [4]INSERM, UMR1287, Gustave Roussy, Villejuif, France; [5]Gustave Roussy, Villejuif, France; [6]Université de Paris, Paris, France; [7]Hamilton Institute, Maynooth University, Co Kildare, Ireland

**Abstract** High-throughput single-cell methods have uncovered substantial heterogeneity in the pool of hematopoietic stem and progenitor cells (HSPCs), but how much instruction is inherited by offspring from their heterogeneous ancestors remains unanswered. Using a method that enables simultaneous determination of common ancestor, division number, and differentiation status of a large collection of single cells, our data revealed that murine cells that derived from a common ancestor had significant similarities in their division progression and differentiation outcomes. Although each family diversifies, the overall collection of cell types observed is composed of homogeneous families. Heterogeneity between families could be explained, in part, by differences in ancestral expression of cell surface markers. Our analyses demonstrate that fate decisions of cells are largely inherited from ancestor cells, indicating the importance of common ancestor effects. These results may have ramifications for bone marrow transplantation and leukemia, where substantial heterogeneity in HSPC behavior is observed.

*For correspondence:
leila.perie@curie.fr

†These authors contributed equally to this work

Competing interests: The authors declare that no competing interests exist.

## Introduction

The hematopoietic system has long since served as a reference model for stem cell biology, with understanding garnered from the study of hematopoietic stem cells (HSCs) successfully transferred to the clinic. In order to maintain blood cell production, rare self-renewing HSCs produce differentiated cells called multi-potent progenitors (MPPs), which proliferate and differentiate through an amplifying cascade of increasingly committed progenitors, ultimately resulting in all mature blood cell types. Underpinning this traditional model is the assumption that the HSC pool is maintained through a process of asymmetric division that results in one HSC and one MPP, while MPPs form a transient cell type that cannot persist indefinitely and must ultimately differentiate.

Recent studies have challenged this theory in multiple distinct directions. It is well-established that HSCs can sequentially reconstitute the blood system of several hosts (*Ross et al., 1982*), leading to the inference that HSCs must be able to maintain themselves (*Morrison and Kimble, 2006*). When observed with time-lapse imaging, self-renewal has been seen to occur through both symmetric and asymmetric cell division (*Wu et al., 2007*; *Brummendorf et al., 1998*; *Ema et al., 2000*; *Punzel et al., 2003*), which can be influenced by extrinsic signals (*Ema et al., 2000*; *Punzel et al., 2003*). Steady-state in situ lineage-tracing studies have also suggested that MPPs are capable of self-renewal (*Sun et al., 2014*; *Busch et al., 2015*). In addition, HSCs have been shown to differentiate without division into megakaryocytes in vitro (*Roch et al., 2015*), and common myeloid,

megakaryocyte, and erythroid progenitors in vivo (*Grinenko et al., 2018*). Together, these findings not only questioned the necessity for HSCs to undergo asymmetric division, but they also queried the explicit link between division and differentiation. Evidence for the multi-potency of HSCs and MPPs has historically derived from in vitro colonies assays and transplantation experiments (*Morrison and Weissman, 1994*; *Osawa et al., 1996*; *Nakahata and Ogawa, 1982*; *Christensen and Weissman, 2001*). Recent single-cell transplantation and cellular barcoding experiments have revealed that only a few HSCs reconstitute all of the blood lineages, with the rest being either restricted in the number of lineages they produce (*Yamamoto et al., 2013*; *Sanjuan-Pla et al., 2013*; *Rodriguez-Fraticelli et al., 2018*; *Carrelha et al., 2018*) or having a bias or imbalance in the proportion of cell types they create (*Dykstra et al., 2007*; *Müller-Sieburg et al., 2004*). As examples of lineage restriction, it has been reported that some HSCs produce only megakaryocytes (*Rodriguez-Fraticelli et al., 2018*; *Carrelha et al., 2018*), while others produce only myeloid cells, megakaryocytes, and erythrocytes (*Yamamoto et al., 2013*). Furthermore, single-cell transplantations have revealed that HSCs are heterogeneous in differentiation and proliferative output (*Benz et al., 2012*; *Müller-Sieburg et al., 2002*; *Sieburg et al., 2011*), and transplantation experiments using populations of HSCs labeled with different fluorescent proteins have suggested that this heterogeneity might be epigenetically determined (*Yu et al., 2017*). Taken together, HSCs have been shown to be a heterogeneous population, where each one of them may be committed to the production of only a few lineages, possibly through lineage priming or externally through instruction from a niche. Similarly, transplanted barcoded MPPs have been reported to produce heterogeneous patterns of restricted cell types (*Naik et al., 2013*), suggesting that lineage restriction may occur early in the hematopoietic tree, in the pool of HSCs and MPPs (*Perié and Duffy, 2016*).

Altogether, it is presently unclear how symmetric and asymmetric division combines with early lineage commitment to generate downstream diversity, and a fundamental question is how much instruction is inherited by offspring from an ancestral HSC or MPP. That matter has not been addressed previously due to technical limitations. Tackling it requires an experimental system that enables the simultaneous identification of cells that are descendent from a common ancestor, the number of divisions that has led to each of them, and their differentiation status. Towards that end, we developed a division-dye multiplex system (*Horton et al., 2018*) for the study of hematopoietic system.

The data from our study revealed that cells that derived from a common ancestor were highly concordant in their division progression and similar in their differentiation pattern. This similarity is primarily propagated through divisions resulting in siblings of the same cell type. These data establish that early lineage commitment can be inherited from individual HSCs and MPPs, and that the resulting diversity of lineages is produced by a heterogeneous collection of cell families that are individually homogeneous. Our data suggests that common ancestor effects are significant and call for a revision of the assumption of independent fate decisions by cells along the hematopoietic tree.

## Results

### High-throughput simultaneous tracking of the common ancestor, number of divisions, and differentiation status of HSPCs

Defining a family as all descendants from an individual marked ancestor cell, we developed a new high-throughput method that simultaneously determines for each cell's family membership, generation (i.e., number of cell divisions), and phenotype (*Horton et al., 2018Figure 1*) in hematopoietic stem cells and progenitors (HSPCs). We focused our investigation on familial division and differentiation in early hematopoietic differentiation to elucidate how symmetric and asymmetric fates combine with early lineage commitment to generate downstream diversity and how much instruction is inherited by offspring from single ancestral HSPC.

To this end, we isolated bone marrow (BM) cells, labeled them with four distinguishable combinations of 5-(and 6)-carboxyfluorescein diacetate succinimidyl ester (CFSE) and CellTrace Violet (CTV), and used fluorescent antibody staining of cell surface markers to determine three HSPC populations, c-Kit$^+$Sca-1$^+$CD150$^+$Flt3$^-$ (SLAM-HSC), c-Kit$^+$Sca-1$^+$CD150$^-$Flt3$^-$ (ST-HSC), and c-Kit$^+$Sca-1$^+$CD150$^-$Flt3$^+$ (MPP) (*Figure 1*). Wells in a 96-well plate were then seeded with four cells of a single ancestral type (SLAM-HSC, ST-HSC, or MPP), one from each of the four CFSE/CTV combinations to

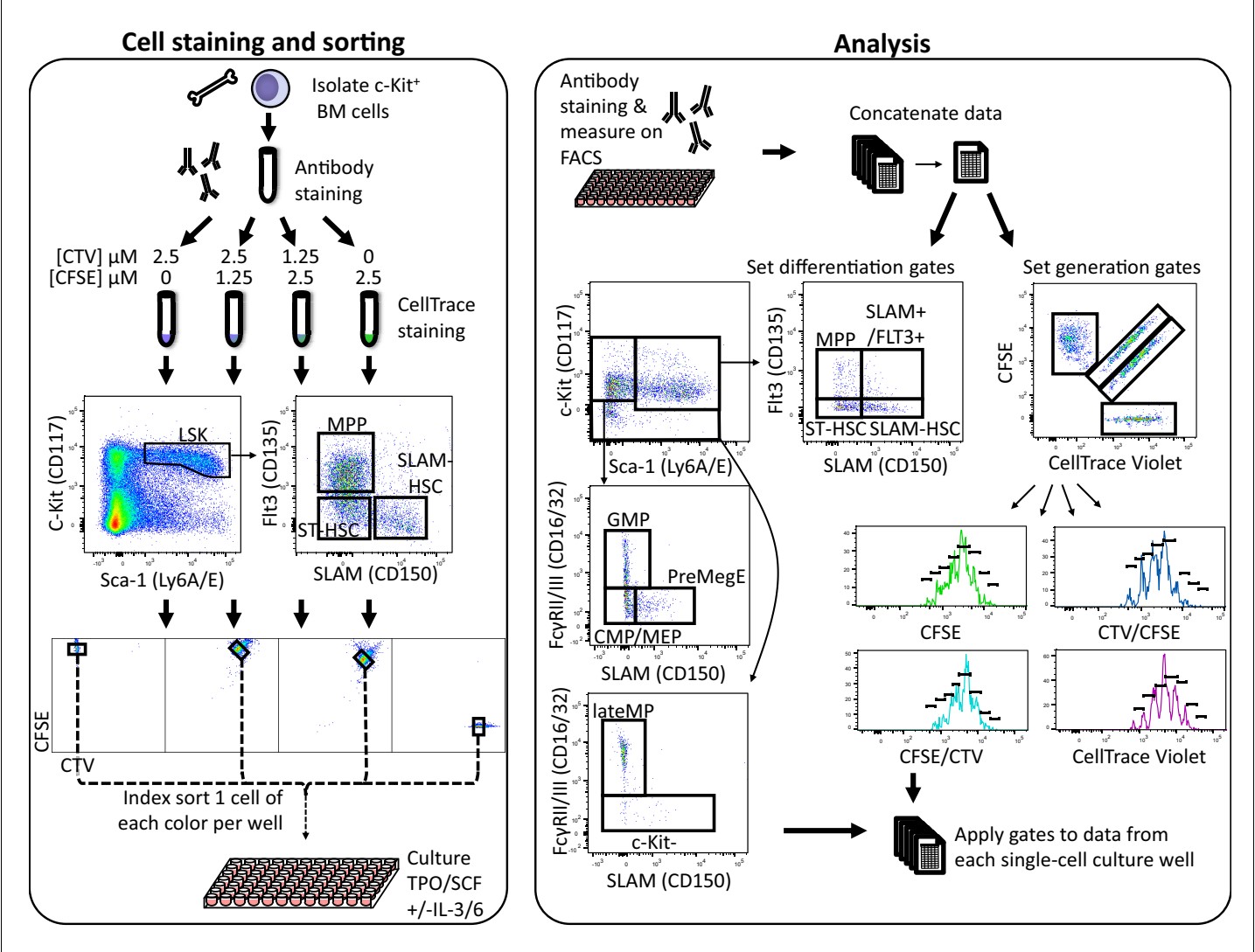

**Figure 1.** High-throughput simultaneous division and differentiation tracking per ancestor. A single-cell suspension was obtained by flushing femurs, tibia, and iliac crests. Cells were stained with fluorescently labeled antibodies for phenotypic identification. After this first antibody staining, the cell suspension was then split into four equal parts, each of which was stained with a distinct 5-(and 6)-carboxyfluorescein diacetate succinimidyl ester (CFSE) and CellTrace Violet (CTV) combination. From each of these CTV/CFSE preparations, a single cell was index sorted into 90 wells of a 96-well plate, resulting in four distinctly CTV/CFSE-colored cells per well. In addition, for each ancestor type, a small bulk of 100 cells of each color combination were sorted into a single well as a control. After 24 or 48 hr of culture, cells were stained with fluorescently labeled antibodies for phenotypic identification and analyzed on a flow cytometer. The data from all wells and the small 100 cell bulks were combined and used to set gates for determination of generation number and phenotypic cell type. Those gates were then applied to the data from each well to obtain lineage, division, and differentiation information for each cell.

The online version of this article includes the following source data for figure 1:

**Source data 1.** Cell type assignment based on cell surface marker.

increase the throughput of the assay, and incubated in one of two classic cytokine cocktails (SCF and TPO, ±IL-3 and IL-6). At 24 or 48 hr, cells were harvested from each well and stained with fluorescent antibodies to determine their phenotypic cell type based on the expression of CD150, Flt3, Sca-1, c-Kit, and CD16/32 (*Figure 1*). By examining each cell's CFSE and CTV profile, its ancestral cell and generation number was determined (*Figure 1* and Materials and methods). By index sorting labeled cells, we could relate downstream familial fate and division to ancestral cell surface expression.

The method allows easily to assess a large collection of single ancestor cells. For each ancestor type, 360 initial cells were sorted for analysis at 24 hr and 240 initial cells for analysis at 48 hr

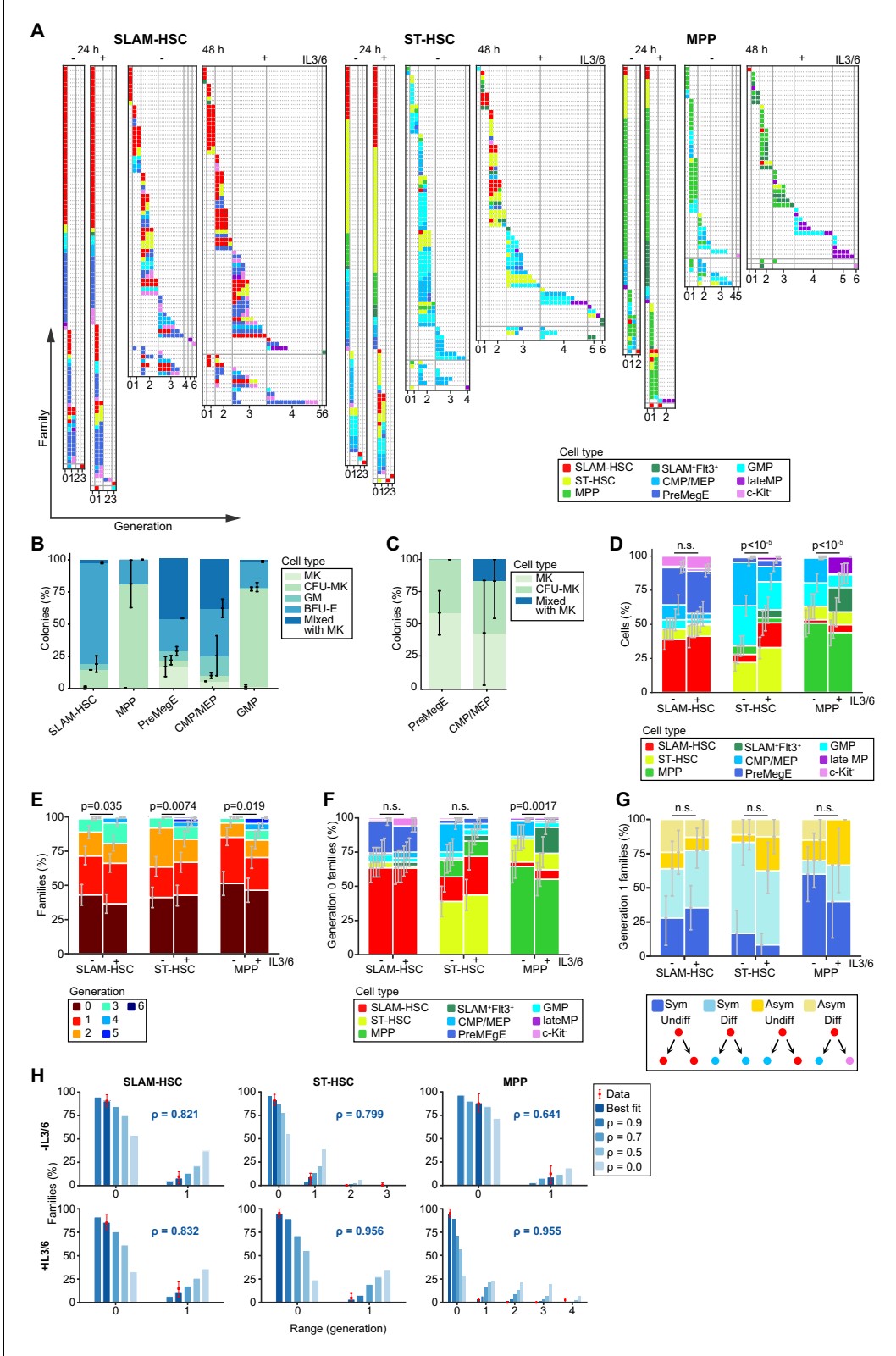

**Figure 2.** Despite population-level heterogeneity, individual hematopoietic stem and progenitor cell (HSPC) families are substantially homogeneous. Plots are fractionated by each ancestor type (SLAM-HSCs, ST-HSCs, and multi-potent progenitors [MPPs]) and cocktail (with and without IL-3 + IL-6, indicated by + and -, respectively). (**A**) Simultaneous visualization of family membership, generation number, and cell type of offspring from initially seeded ancestors harvested at two time points (24 and 48 hr). Each row presents the offspring from a single ancestor. Columns identify the generation

*Figure 2 continued on next page*

*Figure 2 continued*

number of each recovered cell, with their phenotypic cell type indicated by color coding. Rows are sorted in increasing order of the difference between maximum and minimum generations in each ancestor's family (generational range). (B) To test the differentiation pattern of our cell phenotypes after culture, sorted cell types as shown in *Figure 1* were cultured in methylcellulose cultures and evaluated for colony forming units (CFUs) formation for megakaryocytic (MK), granulocytic/monocytic (GM), erythroid burst-forming (BFU-E), or mixed colony formation of all types (CFU-MK, CFU-GM, and BFU-E) with or without MK. Late MP did not produce any colonies in methylcellulose culture (C) The percentages of MK, CFU-MK, and mixed colony formation are indicated from the culture in serum-free fibrin clot. GMP did not produce any colonies as expected. (D) Percentage of each recovered cell type. (E) Distribution of the maximum generation per family, as indicated by color coding. (F) Proportions of recovered cell types for ancestors that have not yet divided. (G) For ancestors that have divided only once and for whom two offspring are recovered, percentage having symmetric and asymmetric fates with and without differentiation. (H) Percentage of families with each generational range. The 48 hr data (red bars) is compared to a mathematical model parameterized by a single coefficient, $\rho$, which encodes the correlation in whether cells in the same generation within a family divide or cease to divide (Materials and methods). Shown is prediction for both the maximum likelihood best-fit value of $\rho$ (value and blue/white bars) and, as a reference, a range of other values of $\rho$ (solid blue bars). For (D–H), error bars indicate 95% confidence intervals calculated via basic bootstrap (Materials and methods). Sample sizes and p-values (from permutation tests, see Materials and methods) from the panels can be found in *Figure 2— source data 1* and *2*.

The online version of this article includes the following source data and figure supplement(s) for figure 2:

**Source data 1.** Number of families per progenitor and condition.
**Source data 2.** Significance values from permutation testing procedures.
**Figure supplement 1.** Cloning efficiency.
**Figure supplement 2.** Fluorescence from antibody staining is not retained during culture.
**Figure supplement 3.** Proportions of recovered cell types for ancestors that have divided only once.

---

(*Figure 2A*). From the 600 seeded SLAM-HSCs, we recovered 358 families (71%) constituting 648 cells, while 343 ST-HSC families (69%) were recovered with 592 cells, and 246 MPP families (49%) with 362 cells (*Figure 2A*). Over all conditions, 27 families (2.8% of recovered families) had cell numbers that could not have originated from a single ancestor, and so were excluded from analysis, illustrating the fidelity of the single-cell sorting.

## Characterization of differentiation outcomes at the population level

At the population level, some offspring underwent no differentiation from their ancestor type while others differentiated. In both culture conditions and for each ancestor type, we obtained a diversity of myeloid cell types ranging from the initial ancestor to c-Kit$^-$ differentiated cells, including c-Kit$^+$-Sca-1$^-$CD16/32$^+$ (GMP), c-Kit$^+$Sca-1$^-$CD150$^-$CD16/32$^-$ (CMP/MEP), c-Kit$^-$CD16/32$^+$ (late myeloid progenitor, late MP), and c-Kit$^-$CD16/32 cell types (*Figure 2A*). We also detected c-Kit$^+$Sca-1$^-$CD16/32CD150$^+$ (PreMegE) cells, previously described as megakaryocyte and erythroid progenitors (*Pronk et al., 2007*). As the phenotypic definitions we used after culture were originally defined on freshly isolated progenitors, we functionally tested the differentiation of the progenitors as defined phenotypically after 48 hr of culture with SCF, TPO, IL-3, and IL-6 using semi-solid cultures (*Figure 2B, C* and *Figure 2—figure supplement 1*) and observed a similar differentiation outcome as previously published (*Pronk et al., 2007*).

Although all hematopoietic cells have been reported to go through a Flt3 expressing stage (*Boyer et al., 2011*), we found that no offspring of SLAM-HSCs, and those of very few ST-HSCs, differentiated into Flt3 expressing MPPs in our culture conditions (*Figure 2D*). Some ST-HSC and MPP acquired CD150 expression after culture, making them resemble SLAM-HSCs, but this was not further investigated. This CD150 expression cannot be due to residual fluorescence from antibodies used for cell sorting as no fluorescence was measured for any of the markers in cells kept for 24 hr in culture after sorting without further antibodies staining (*Figure 2—figure supplement 2*). The addition of IL-3 and IL-6 had no impact on the pattern of cell types produced by SLAM-HSCs (*Figure 2D*). It did, however, change the pattern of cell types produced by ST-HSCs and MPPs in a statistically significant way as determined by a permutation test (Materials and methods), with an increased number of early progenitors as opposed to more differentiated progenitors as previously shown (*Lui et al., 2014*).

## Heterogeneity between HSPC families in division and differentiation

At the level of individual families, we observed substantial heterogeneity in division history (*Figure 2E*) from families that did not proliferate to those with cells that had undergone six divisions. Consistent with earlier observations (*Roch et al., 2017*), after 24 hr most cells either remained undivided or had divided once, with a few cells having undergone two or more divisions (*Figure 2A*). At 48 hr, over 90% of the cells had undergone at least one division. For all three sorted ancestor cell types, we found that addition of IL-3 and IL-6 led to a statistically significant increase in proliferation (*Figure 2E*), as previously described (*Domen and Weissman, 2000*; *Bordeaux-Rego et al., 2010*). Exploring the relationship between division and differentiation, in the culture without IL-3 and IL-6, the proportion of undivided ancestors that differentiated were 36.5% from SLAM-HSCs, 61.1% from ST-HSCs, and 35.6% from MPPs (*Figure 2F*), which was in agreement with previous reports (*Roch et al., 2015*; *Grinenko et al., 2018*). The addition of IL-3 and IL-6 did not drastically change those values. SLAM-HSCs preferentially differentiated without dividing into PreMegEs, as reported previously (*Grinenko et al., 2018*). On comparing the surface marker expression of differentiated and not-differentiated non-divided cells (*Figure 2—figure supplement 3*), Sca-1$^{high}$ SLAM-HSCs were more likely not to differentiate, in agreement with a previous report (*Schulte et al., 2015*). The addition of IL-3 and IL-6 only significantly impacted the differentiation pattern of the progeny of MPPs. These results show that families are heterogeneous in their division pattern, and that a non-negligible fraction of ancestors differentiates without dividing.

## Both symmetric and asymmetric fate occurred within families after the first division

Our experimental system can capture a large number of siblings after a single division, enabling the quantification of symmetric versus asymmetric fates. We defined four distinct types of symmetric or asymmetric fates depending on whether the offspring included a differentiated cell (*Figure 2G*). A symmetric undifferentiated fate produces two cells of the ancestor type. An asymmetric undifferentiated fate, which would be the classically defined asymmetric fate in the stem cell community, produces one cell of the ancestor type and one differentiated cell. Similarly, a symmetric differentiated fate produces two cells of the same differentiated type, and an asymmetric differentiated fate produces two cells of the distinct differentiated type. Note that an asymmetric fate cannot be distinguished from a symmetric fate followed by differentiation without division of one of the daughter cells. Pooling data over ancestor types, 70.7% of the cells had a symmetric fate after their first division, with the fate of MPPs being mostly symmetric undifferentiated (51.4%), and of ST-HSC being mostly symmetric differentiated (59.5%). SLAM-HSCs self-renewed with symmetric undifferentiated fates (32.1%), but also with asymmetric undifferentiated fates (10.7%). Asymmetric fates occurred for 28.6% of SLAM-HSCs, 28.6% of ST-HSCs, and 31.4% of MPPs, consistent with previous reports (*Suda et al., 1984*). No statistical difference was found between ancestors cultured with or without IL-3 and IL-6. The pattern of cell types produced after one division (*Figure 2* and *Figure 2—figure supplement 3B*) was similar to the pattern including all divisions (*Figure 2D*), suggesting that the diversity of cell types can be produced by a heterogeneous collection of cell families through symmetric fates.

## HSPC family members were concordant in division

To investigate familial effects on division progression, we examined the generation numbers of cells derived from single ancestors. We found that families are highly concordant, with 81% of the 223 families that divided more than once having all of their cells in the same generation (range = 0, *Figure 2H*), and only four of those families (1.8%) containing cells that were more than one generation apart (range >1, *Figure 2H*). As not all cells are necessarily recovered from wells and sampling effects could potentially make families look more concordant, a mathematical model that accounts for that sampling was fitted to the 48 hr data to estimate the correlation in division progression decisions among cells within a family (Materials and methods). High correlation coefficients (70–90%) (*Figure 2H*) resulted in the best fit to the measured familial ranges, with the exception of MPPs cultured in medium without IL-3 and IL-6 for which no reliable estimate could be made. For guidance, the pattern of range values from the model for different correlation coefficients is also illustrated in

*Figure 2H*. Thus, this analysis establishes that division progression is highly concordant within SLAM-HSC, ST-HSC, and MPP families, while being heterogeneous between them.

## Differentiation occurred though a diversity of paths and progressed with division

Although differentiation without division was observed, in general differentiation progressed in tandem with division (*Figure 3A*), as published previously (*Upadhaya et al., 2018*). To visualize changes in cell surface marker expression, we used the Uniform Manifold Approximation and Projection (UMAP) algorithm (*McInnes and Healy, 2018*) on the combined surface marker expressions of all cells obtained at both 24 and 48 hr (*Figure 3B* and *Figure 3—figure supplement 1*). When we mapped cell types determined by traditional gating onto the UMAP (*Figure 3C*), a smooth transition was observed from SLAM-HSCs at the bottom, with further differentiated cells towards the top, indicating a gradual transition from one cell type to another. GMPs and c-Kit$^-$Sca-1$^-$CD16/32$^+$ (myeloid progenitors [MPs]) appear on the top left, and CMP/MEP and PreMegE on the top right. More numerous GMPs, MPs, and CMP/MEPs were seen at the top of the UMAP at 48 hr than 24 hr, suggesting that it takes between 24 and 48 hr to fully differentiate into Sca1$^-$ progenitors. In addition, all three ancestral cell types remain present at 48 hr, demonstrating, in particular, that HSCs can remain in an undifferentiated state for the duration of the experiment, even if their offspring experience three rounds of division. On plotting the generation numbers of offspring from each ancestor cell type on the UMAP (*Figure 3D* and *Figure 3—figure supplement 2*), SLAM-HSCs appeared to be primed towards the production of CMP/MEPs and PreMegEs while still generating some ST-HSCs, whereas MPPs were more primed towards GMPs, and ST-HSCs showed a more even distribution between the two lineages. Differentiation without proliferation appeared as dark red dots outside of the regions of the sorted ancestor cells, and self-renewal divisions as red, orange, and blue dots in the region of ancestor cells. PreMegEs were observed to be generated without division, as well as in 1–3 divisions from SLAM-HSCs.

## HSPC family members displayed similar differentiation outcome

Descendants from a common ancestor were not only highly concordant in their generation numbers, but they also exhibited significant similarity in differentiation outcome. At 24 hr, most families were composed of only one cell type (*Figure 4A*), but at 48 hr more families produced several cell types (*Figure 4B*), indicating that downstream asymmetries in the fate occur after a largely symmetric first division (*Figure 2G*). Permutation tests on phenotypically defined cell types revealed that families exhibited significantly more similarity than would be expected if there was no family component, both at 24 hr (*Figure 4A*) and at 48 hr (*Figure 4B*). This within-family similarity in fate is visible by the co-localization of cells from the same family on the UMAP (*Figure 4C*). Thus, SLAM-HSC, ST-HSC, and MPP families are highly concordant in division and share similar differentiation outcomes in vitro, while population-level diversity in proliferation and cell types arises from heterogeneity across families.

The within-family homogeneity in division and differentiation could be intrinsically present in ancestor cells or extrinsically instructed by cytokines in the cocktail. Comparison of the median fluorescent intensities of markers from ancestors during sort with those obtained from daughters at each of the two time points (*Figure 4—figure supplement 1A*) revealed clear correlation between the two, supporting the hypothesis that ancestor surface expression markers were instructive in the within-family homogeneity in division progression and differentiation. To further investigate that hypothesis, we first explored the relationship between cell surface expression on the ancestor and division progression. Rank ordering ancestors from the least to greatest expression level for a given marker (*Figure 4D*, *Figure 4—figure supplement 1B*), the cumulative sum of the maximum division of their offspring would be expected to fall near the diagonal if there were no relationships between an ancestral expression level and division progression. If there was a negative relationship, where low expression of a given marker on the ancestor corresponded to more division progression, the cumulative sum of the maximum division would be expected to initially overshoot above the diagonal. The contrary would happen if there was a positive relationship. The statistical significance of divergence from the diagonal was tested using Jonckheere's trend test (*Figure 4—source data 1*). Across both cocktails and time points, only cell surface markers on ancestral SLAM-HSCs were

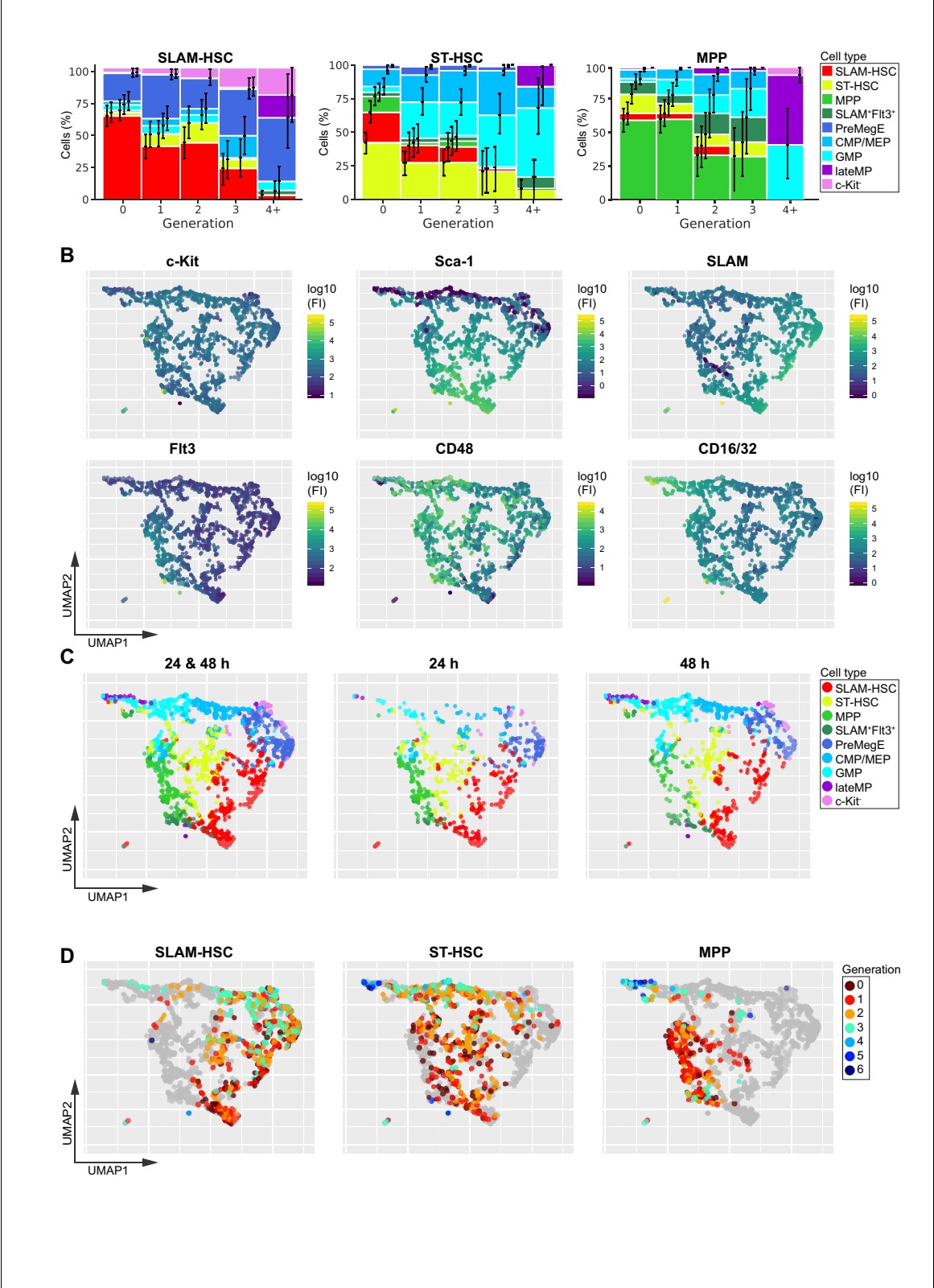

**Figure 3.** Differentiation and division progress in tandem. (**A**) Percentage of cells at each differentiation stage for each generation for each ancestor type. Error bars indicate basic bootstrap 95% confidence intervals (Materials and methods). Sample sizes for this panel can be found in source data. (**B**) The Uniform Manifold Approximation and Projection (UMAP) algorithm was applied to the phenotypic data pooled from all time points, conditions, and ancestor cell types. Each cell is projected into the UMAP coordinates and color-coded according to the log of their fluorescence intensity for c-Kit,

*Figure 3 continued on next page*

Figure 3 continued

Sca-1, CD150, Flt3, CD48, and CD16/32 at the time of analysis (see also *Figure 3—figure supplement 2*; and for each ancestor cell type plotted separately on the UMAP, see *Figure 4—figure supplement 1*). (C) Projection of traditionally gated data onto the UMAP, pooled data (left), 24 hr (middle), and 48 hr (right). (D) Projection of cell generation number data onto the UMAP for each ancestor type (see also *Figure 3—figure supplement 3*). Sample sizes for panel (A) can be found in *Figure 3—source data 1*.

The online version of this article includes the following source data and figure supplement(s) for figure 3:

**Source data 1.** Number of cells per generation from a given progenitor.
**Figure supplement 1.** Additional information plotted onto the differentiation of Uniform Manifold Approximation and Projection (UMAP).
**Figure supplement 2.** Generation numbers projected onto the Uniform Manifold Approximation and Projection, fractionated by ancestor type and time point.
**Figure supplement 3.** Per ancestor type fluorescence intensities during sort projected on the Uniform Manifold Approximation and Projection.

consistently instructive for division progression. CD48 correlated positively, with its strongest effect at 24 hr, and Sca-1 correlated negatively, while at 48 hr c-Kit correlated positively (*Figure 4D*, *Figure 4—figure supplement 1B*, and *Figure 4—source data 1*). Notably, for both ST-HSCs and MPPs, even though the family data clearly indicate that there is a familial component to division progression (*Figure 2H*), none of the phenotypic markers exhibited strong correlation (*Figure 4—figure supplement 1B*), indicating the need to identify other markers.

## Ancestral phenotype correlated with familial differentiation outcome

We then explored the relationship between fluorescence intensity of ancestral marker expression and familial differentiation (*Figure 4E*, *Figure 4—figure supplement 2*, and *Figure 4—source data 2* and *3*; for a summary of findings in table format, see *Figure 4—source data 4*). For SLAM-HSCs, at 24 and 48 hr in both cocktails, Sca-1 expression provided a strong positive correlation to self-renewal and a negative one to production of PreMegE. At 48 hr, c-Kit presented the inverse dependency to Sca-1. At 24 and 48 hr, CD48 correlated negatively to self-renewal and positively to production of PreMegE when IL-3 and IL-6 are added. As cocktail composition did not have a major impact on the relationship between familial fate and ancestral expression, these results were suggestive that c-Kit and Sca-1 expression levels of SLAM-HSCs act as intrinsic markers for both familial progression and differentiation with high Sca-1 expression and low c-Kit expression leading to less division (*Wilson et al., 2015*; *Grinenko et al., 2014*; *Morcos et al., 2017*) and less differentiation (*Shin et al., 2014*), and potentially resulting in better engraftment (*Wilson et al., 2015*; *Grinenko et al., 2014*; *Shin et al., 2014*). While low ancestral CD48 expression level has been reported to result in less division (*Pietras et al., 2015*; *Akinduro et al., 2018*), our data indicates its relationship to differentiation is dependent on extrinsic signals.

For ST-HSCs and MPPs, we found little evidence of correlation of ancestral expression to division progression or self-renewal, but the same was not true of differentiation. For ST-HSCs, the ancestral level of CD48 and Sca-1 consistently correlated negatively and positively, respectively, with de-differentiation to SLAM-HSC in the cocktail with IL-3 and IL-6 at both time points (*Figures 4E* and *Figure 4—figure supplement 2*). Differentiation to GMP, which occurred only in 48 hr data, correlated positively and negatively with the ancestral level of CD48 and Sca-1, respectively (*Figure 4E*; *Morcos et al., 2017*). Therefore, differentiation to GMP from ST-HSC was dependent on the parental level of CD48 and Sca-1, whereas the de-differentiation to SLAM-HSC is dependent on both extrinsic factors (IL-3 and IL-6) and the intrinsic ancestral level of CD48 and Sca-1. The differentiation from MPPs to GMPs that was observed to occur by 48 hr correlated negatively with Sca-1 ancestral expression (*Morcos et al., 2017*) in both cocktails. It also negatively correlated with Flt3 ancestral expression, but only in the cocktail without IL-3 and IL-6 (*Figure 4E*). In the presence of IL-3 and IL-6, instead, differentiation from MPP to GMP positively correlated with c-Kit ancestral expression. Differentiation from MPP to CMP/MEP occurred only at 48 hr in the cocktail without IL-3 and IL-6, and then correlated positively with c-Kit (*Figure 4E*). Thus, differentiation to GMP from MPP is dependent on the intrinsic ancestral level of Sca-1, whereas the differentiation to CMP/MEP is dependent on both extrinsic factors (IL-3 and IL-6) and the intrinsic ancestral level of c-Kit. Overall, the concordance in division and similarity in fate within families is partially explained by the surface expression marker used to phenotype ancestors, but both intrinsic and extrinsic factors act to direct familial fate.

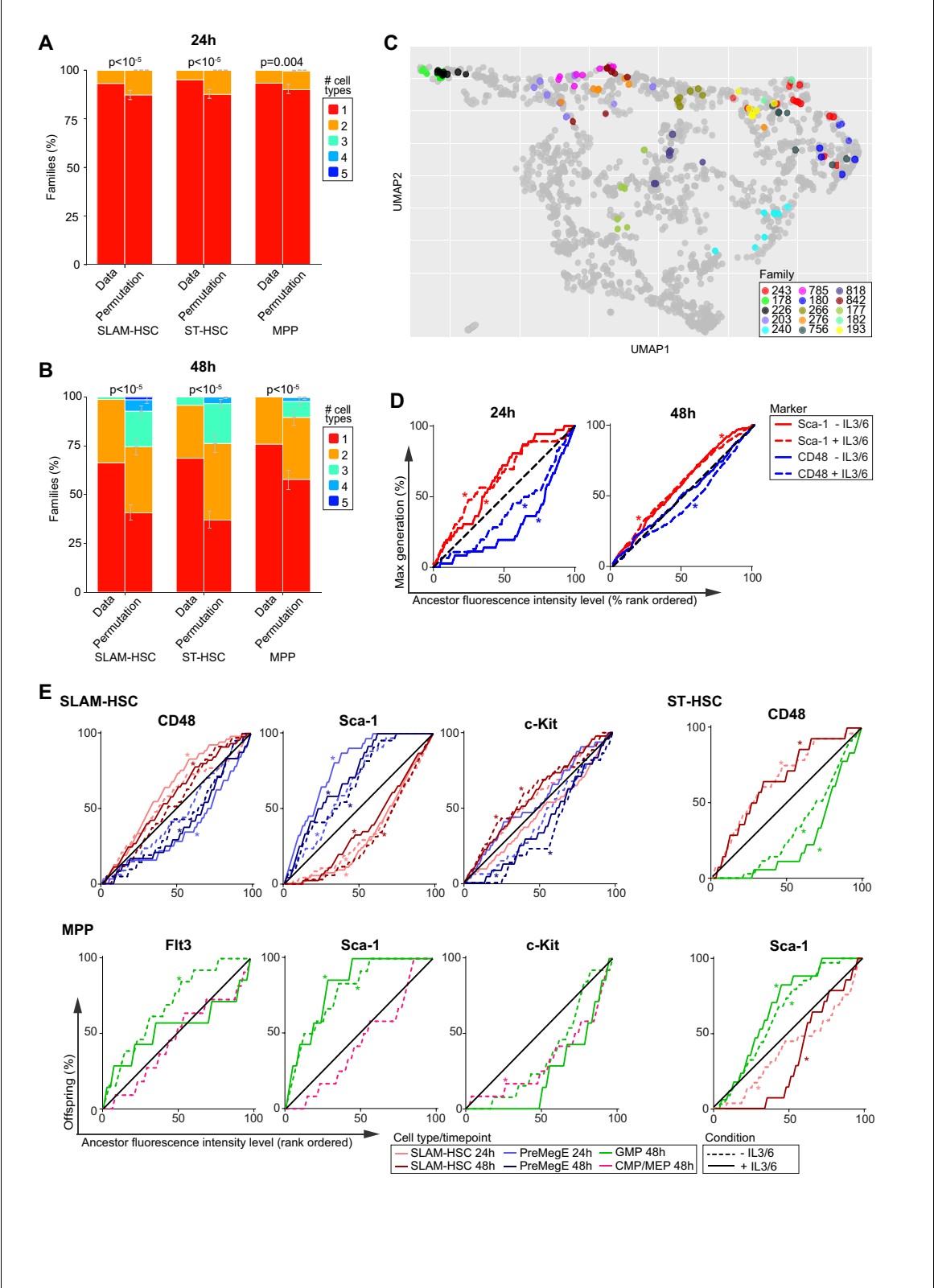

**Figure 4.** Families are highly concordant in differentiation. (A, B) Number of cell types per family in the observed data compared with the average of 250,000 permutations of the data at 24 hr (A) and 48 hr (B). Error bars indicate 95% confidence intervals based on permutations (see Materials and methods). (C) Cells from the 15 families with the largest number of cells are color-coded by family and projected onto the Uniform Manifold Approximation and Projection in *Figure 3*. (D) The cumulative percentage of the maximum division of offspring from ancestor SLAM-HSCs

*Figure 4 continued*

rank-ordered by their expression level (fluorescence intensity) of CD48 (blue) or Sca-1 (red) during sort. (**E**) The cumulative percentage of offspring, presenting a given cell type, from ancestor cells rank-ordered by increasing cell surface marker expression. * indicates a significant deviation from the diagonal (black) as determined by Jonckheere's trend test (p-values in *Figure 4—source data 1*). Sample sizes for all panels can be found in *Figure 4—source data 2*.

The online version of this article includes the following source data and figure supplement(s) for figure 4:

**Source data 1.** Significance values from the Jonckheere's rend test between expression levels at sort and offspring maximum generation as shown in *Figure 4D*.

**Source data 2.** 95% confidence intervals of Spearman r and p-values of correlations.

**Source data 3.** Significance values from the Jonckheere's trend test between expression levels at sort and presence of a given cell type among offspring.

**Source data 4.** Summary of significant findings as shown in *Figure 4E*.

**Figure supplement 1.** fluorescence intensity correlation within families and correlation with division for additional markers and ancestor types.

**Figure supplement 2.** The cumulative sum of the offspring presenting a given cell type from ancestor cells rank-ordered by cell surface marker expression as in *Figure 4E*, without normalizing the offspring count, and plotted for each ancestor cell type, time point, culture condition, and marker.

## Discussion

We developed a high-throughput method that enables simultaneously determination of common ancestor, generation, and differentiation status of a large collection of single cells. Its use with HSPCs revealed that despite substantial population-level heterogeneity amongst offspring cells derived from a single ancestor are highly concordant in their division progression and exhibit familial effects on differentiation. The restriction in differentiated cell types within each family is propagated primarily through symmetric first divisions. Although each family is composed of several cell types, the overall collection of cell types observed in a population is composed of homogeneous families from heterogeneous ancestors. This finding opens new avenues and challenges for the hematopoietic field. The generation of a diversity of cell types is presently assumed to result from a diversification within every family, and methods for inferring differentiation trajectories using single-cell RNA sequencing data from snapshot data assume that cells all behave independently (*Trapnell et al., 2014*; *Bendall et al., 2014*). Consistent with previous observations of early lineage priming (*Müller-Sieburg et al., 2002*; *Perié and Duffy, 2016*; *Paul et al., 2016*; *Hoppe et al., 2016*), our findings establish that familial dependencies that are currently unmeasured exist within the population and call for a revision of the assumption of independent fate decision by cells along the hematopoietic tree. Ancestral cell surface expression of markers used for phenotyping serves as correlates that partially predict some of these familial properties, but, in particular, a correlate that explains the highly heritable division progression of ST-HSC and MPP families is not contained within them. It is also the case that extrinsic properties such as cytokine signaling can play an instructive role, altering and reshaping the observed familial effects.

As HSPCs are cultured before BM transplantation in gene therapy, our results indicate that the broad range of engraftment and proliferation capacities of HSPCs could be consequences of the heterogeneity in their engrafted families. That suggests that altered culture conditions might reduce or enhance heterogeneity between families and possibly improve transplantation outcomes if this leads to more self-renewal divisions. Indeed, changing the composition of the population of committed HSPC might be a mechanism to directly alter the balance of lineage production, with therapeutic applications that could benefit the treatment of leukemia and genetic immune disorders.

## Materials and methods

### Mice and cell isolation

All the experimental procedures were approved by the local ethics committee (Comité d'Ethique en expérimentation animale de l'Institut Curie) under approval number DAP 2016 006. BM cells were obtained from wild-type C57BL/6 of 8–16 weeks of age by bone flushing of femur tibia and iliac crest. BM cells were MACS enriched for c-Kit$^+$ cells using CD117 MicroBeads Ultrapure (Miltenyi Biotec cat #130-091-224) according to the manufacturer's protocol.

Division tracking and surface marker labeling of HSPC c-Kit-enriched BM cells were stained with CD135 (Flt3) PE (eBiosciences 12-1351-82), Sca-1 PE-CF594 (BD Biosciences, 562730), CD117 (c-Kit)

APC (Biolegend 105812), CD150 (SLAM) PC7 (Biolegend 115914), and CD48 APC-Cy7 (Biolegend 103432) in RPMI1640 supplemented with 10% FCS. Subsequently, cells were stained in PBS with either 2.5 µM CellTrace CFSE (ThermoFisher Scientific C34554), 2.5 µM CTV (ThermoFisher Scientific C34557), and 2.5 µM CFSE together with 1.25 µM CTV or 2.5 µM CTV together with 1.25 µM CFSE (see *Figure 1A*) as adapted from.

Single c-Kit$^+$Sca-1$^+$CD150$^+$Flt3$^-$ (SLAM-HSC), c-Kit$^+$Sca-1$^+$CD150$^-$Flt3$^-$ (ST-HSC), and c-Kit$^+$Sca-1$^+$CD150$^-$/Flt3$^+$ (MPP) were sorted directly into U-bottom 96-well plates containing cell culture media using an Aria III cell sorter (BD Biosciences). For each cell type, we sorted four single cells, one for each of the CellTrace stain combinations, into each well. Sorting four ancestor cells per well is a critical step in the method to ensure that at time of analysis there are enough cells in the well, which could not be obtained when sorting one ancestor cell per well. In total, 30 wells (120 single cells) were sorted per cell type per plate, with three replicates for analysis at 24 hr and two replicates for analysis at 48 hr. In addition, we sorted 100 cells of each cell type into one well for both culture conditions in order to collect enough events for reliable gate definition for cell type and generation assignment. During the sort of single cells, fluorescence intensities of each surface marker were recorded using the index sorting function.

## In vitro cell culture

Cells were cultured at 37°C under 5% $CO_2$ in 100 µl of StemSpan serum-free expansion medium (Stemcell Technologies 9650) supplemented with 50 ng/ml murine recombinant thrombopoietin (TPO, Sigma-Aldrich SRP3236-10UG) and 100 ng/ml stem cell factor (SCF) or 50 ng/ml TPO, 100 ng/ml SCF, 20 ng/ml IL-3, and 100 ng/ml IL-6 (*Ema et al., 2000*; *Roch et al., 2017*).

## Division and expression marker analysis of cell progeny

After 24 or 48 hr of incubation, cells in each well were stained as for sorting except for the use of CD48 BUV395 (BD Biosciences 740236), Sca-1 APC-Cy7 (Biolegend 108125), and CD16/32 BV711 (BD Biosciences 101337). Cells from each well were analyzed at 4°C using a ZE5 Flow cytometer (Bio-Rad) with a recovery estimate of circa 70% per well (beads-based estimate, data not shown).

## Cell type and generation assignment

For data analysis of FACS data, we pooled all the data from a single experiment using the concatenate function in FlowJo (FlowJo, LLC version 10.4.2). For cell type assignment, gates were set on concatenated data of both single-cell and bulk sorted samples and then applied to the single-cell data (*Figure 1*). Cells were separated from debris by their forward and size scatter (FSC/SSC) profile and assigned to a cell type (see *Figure 1—source data 1*). The generation (i.e., the number of divisions since labeling) of cells was determined on histograms of CellTrace dye fluorescence in FlowJo. For cells stained with both CFSE and CTV, we rotated the CTV/CFSE coordinates, on a logarithmic scale, by 45° degrees anticlockwise so that division dilution proceeded in parallel to the horizontal. That is, with $x$ and $y$ denoting the coordinates of CTV and CFSE levels, the histogram was calculated over a new x-axis coordinate

$$x' = \frac{\sqrt{2}(\ln x + \ln y)}{2}. \tag{1}$$

Generation gating was then determined based on the florescence histogram on the new x'-axis on the merged data of wells from the same experiment.

## Data visualization by UMAP

UMAP (*McInnes and Healy, 2018*) was performed on arcsinh(x/100) transformed fluorescence intensity values of surface expression markers from all experiments using the R implementation in the UMAP package (version 0.2.0.0) with default parameters. The UMAP output was visualized using the ggplot two package (version 3.0.0) in R (version 3.4.3).

## Progenitor assays in semi-solid cultures

SLAM-HSC (150–200 cells), MPP (700–1500 cells), CMP/MEP (1000 cells), GMP (300–1000), PreMegE (1000 cells), and late MP (5000 cells) were plated in duplicate or triplicate in methylcellulose

MethoCult 32/34 (Stemcell Technologies) with 10 ng/ml TPO (a generous gift from Kirin, Tokyo, Japan), 1 U/ml EPO (PreproTech), 10 ng/ml IL-3 (Miltenyi Biotec), 10 ng/ml IL-6 (Miltenyi Biotec), 100 ng/ml SCF (PreproTech), and 20 ng/ml G-CSF (Miltenyi Biotec). Colonies derived from erythroid progenitors (colony forming unit-erythroid [CFU-E]) were counted after 2 days, but no CFU-E was detected in any of the cell populations tested. Colonies derived from erythroid progenitors (burst forming unit-erythroid [BFU-E]), granulo-monocytic (colony forming unit-granulocyte macrophage [CFU-GM]), and multilineage colonies (mixed) progenitors were counted after 9 days. For megakaryocytic progenitor (CFU-MK) assay, SLAM-HSC (150–200 cells), MPP (2000 cells), CMP/MEP (2000 cells), GMP (2000), PreMegE (2000 cells), and late MP (5000 cells) were plated in triplicate in serum-free fibrin clot assays with SCF, IL-6, and TPO. MKs and CFU-MKs were evaluated at day 7 by acetylcholinesterase staining.

## Confidence intervals

The confidence intervals at 95% level shown in *Figures 2B–F* and *3D* were calculated via basic bootstrap (*Davison and Hinkley, 1997*) with 250,000 bootstrap datasets. Following this procedure, each bootstrap dataset is constructed by sampling with replacement as many cellular families (*Figures 2B–H* and *3D*) as were in the original data. The distribution of the statistics, each calculated from one bootstrap dataset, then provided a reference from which the confidence interval was derived. Formally, given the statistic $\theta$ calculated from the original data, and $\theta^*_{(0.025)}$ and $\theta^*_{(0.975)}$ the 0.025 and 0.975 percentiles, respectively, derived from the bootstrapped distribution, the confidence interval of $\theta$ at 95% level was calculated as follows:

$$CI(95\%) = \left( 2\theta - \theta^*_{(0.975)}, 2\theta - \theta^*_{(0.025)} \right)$$

## Statistical testing framework

To perform the statistical analysis, we adapted the permutation test (*Lehmann and Romano, 2006*) framework proposed in *Horton et al., 2018*. This framework was preferred over classical statistical tests as their assumptions were violated by the presence of familial dependencies in the data.

The objective of this framework was to challenge the hypothesis of independence between one or more variables in the data. For example, to test if the differentiation pattern was changed by culture conditions in *Figure 2D*, we challenged the null hypothesis that differentiation pattern per ancestor type (e.g., SLAM-HSC) was independent of culture conditions. If that null hypothesis held true, then the pattern of differentiation would not statistically change on swapping families between the culture conditions. Thus, the first step in the procedure consists in computing a statistic for the measured data that captures a key characteristic related to the variables to be tested. In this example, we chose the statistic to be the G-test statistic (or G-value) for contingency tables (*Lehmann and Romano, 2006*); therefore, the differentiation pattern data was transformed into the cellular frequencies from each cell type (columns) for each culture condition (rows). The second step is to perform randomization of the data, the permutation, that will be compared to the measured data. Each randomly selected permutation captures how the data would look if the differentiation pattern and the culture condition were independently assigned. Indeed, if these two variables were independent, we could shuffle cellular families between culture conditions and the composition of the resulting permuted dataset would be statistically similar to the original measured data. If one shuffled cells instead of families, then any familial dependence of cells would break down and so interfere with the testing of the independency between the differentiation pattern and the culture condition. The ability to manage familial dependencies is the reason why this statistical framework is well suited to these data. In *Figure 2D*, cellular families derived from the same ancestor type were permuted between the two culture conditions 250,000 times, and the G-value was then computed for each permuted dataset. Finally, the proportion of the G-values of the permuted datasets that were as, or more, extreme than the G-value from the original dataset determined the p-value of the hypothesis test. This in turn indicated whether the differentiation pattern significantly varied with the culture condition. In general, for each test performed in this paper, a test statistic and data permutation class must be defined to characterize the hypothesis to be challenged and to compute the p-value. Below, a more formal explanation is provided, followed by a paragraph with a description of the statistics and the permutation strategies specifically used throughout this work.

In more mathematical terms, a typical example of permutation testing proceeds in the following manner. A null hypothesis concerning the independence of the data, $D$, on one or more variables is first determined. Then, a collection, $Q$, of permutations of the data is identified such that, under the null hypothesis, the permuted data $D^\pi$, for any $\pi \in Q$, is equal in distribution to $D$. In this way, for any a real-valued statistic $T$ of the data, $T(D)$ and $T(D^\pi)$ are equal in distribution given the data $D$. Therefore, the distribution of $T(D)$, and its associated p-values, can be approximated by the distribution of $\{T(D^{\pi_1}), \ldots, T(D^{\pi_B})\}$, which is obtained by sampling a large number $B$ of permutations $\pi_i$ from $Q$, for $i = 1, \ldots, B$. Of note, the statistic $T$ should be chosen to present good sensitivity with respect to the departure of the data from the null hypothesis, a property often exhibited by classical statistics.

To further clarify the framework described above, we make explicit how the null hypothesis of independence between culture condition and differentiation pattern was challenged in **Figure 2D**. Under this hypothesis, the frequencies of cell types from the two culture conditions each from a different cell culture were equal in distribution. In particular, under the null hypothesis the distribution of the data in each culture condition would not change upon the shuffling of cellular families between culture conditions, which identifies a suitable set of permutations $Q$. As the variables to be tested were either discrete or categorical, the independence of cell-type frequency from the culture condition was tested selecting $T$ to be the G-test statistic for contingency tables (**Lehmann and Romano, 2006**). Following this rationale, the same choice for the set of permutations, $Q$, and the test statistics, $T$, was made to challenge the hypotheses of independence between culture condition and the other discrete variables: maximum division number per family (**Figure 2E**), differentiation pattern without division (**Figure 2F**), and pattern of first division (**Figure 2G**).

In **Figure 4B**, we sought to challenge the null hypothesis that differentiation diversity among cells from the same ancestor type was independent of familial membership, effectively testing whether a cell's familial membership was independent of its type. Under this null hypothesis, the naïve assumption would be to define $Q$ as the set of permutations that swap cells between or within families, but, as cell type appeared to correlate with cell generation (**Figure 3D**), permuting cells with different division number would return a dataset $D^\pi$ that is not equal in distribution to $D$, the original. Leveraging the flexibility of the testing framework, it sufficed to instead restrict the set $Q$ to be permutations that leave the generations of cells unaltered, effectively solely swapping cells (between or within families) having the same division number. For this test, $T$ was set as the average number of cell types per family since this statistic is expected to decrease under the alternative hypothesis that cells with a common ancestor diversify into a smaller collection of cell types.

Finally, we tested whether the ancestor's expression levels were independent of an ordinal variable of its offspring: division (**Figure 4D**) and differentiation pattern (**Figure 4E**). Under each null hypothesis, $Q$ was defined as the set of permutations of families amongst ancestors, which embodies the assumption that a family is assigned independently at random to an ancestor. To assess such null hypothesis when compared against the alternative that the families ranked by their ancestors' expression levels established a trend (either increasing or decreasing) in the other familial variable, **Jonckheere's trend test** was chosen as the test statistic $T$.

## Statistical testing formulae

To challenge the null hypotheses that differentiation was independent of the culture condition, using the data underlying **Figure 2D** we compared the population proportion per cell type. For notational purposes, the data were represented as a sequence $D = (\tau_i, c_i, s(c_i))_{i=1}^N$ of $N$ cells, where the $i$th cell was identified by cell type $\tau_i$, family $c_i$, and culture condition of the family $s(c_i)$. To assess the independence of cell types $J = \{\tau_i, i = 1, \ldots, N\}$ from partition labels $l \in \{1, \ldots, L\}$ (relative to culture condition), the statistic $T$ of the data $D$ was defined as the log-likelihood statistic of the G-test for the contingency table $O$, such that $O_{jl} = \sum_{i=1}^N \chi(\tau_i = j, s(c_i) = l)$ with $\chi(A) = 1$ if the event $A$ holds true and 0 otherwise. The G-test statistic is classically used for the testing of independence between two sets of categories ($J$ and $\{1, \ldots, L\}$) partitioning the data counts. Therefore,

$$T(D) = 2 \sum_{l=1}^L \sum_j O_{jl} \, In\left(\frac{O_{jl}N}{E_{jl}}\right) \qquad (2)$$

where $E_{jl} = \left( \sum_{i \in J} O_{il} \right) \left( \sum_{i=1}^{L} O_{ji} \right)$.

Under the null hypothesis that differentiation was not impacted by culture condition, $D$ is equally likely as a dataset $D^{\pi} = (\tau_i, \pi(c_i), s(\pi(c_i)))_{i=1}^{N}$ transformed by the action of any permutation $\pi \in Q$ of the set of family labels $\{c_i, i = 1, \ldots, N\}$. As a consequence, using Monte Carlo approximation we estimated the p-value for the right-tailed test as

$$\hat{p}_B^r = \frac{1 + \sum_{i=1}^{B} \chi(T(D) \leq T(D^{\pi_i}))}{1 + B}, \tag{3}$$

where $B = 250{,}000$ and $\pi_1, \ldots, \pi_B$ were uniformly and independently sampled from $Q$.

To challenge the null hypotheses that familial division was independent of the culture condition, for the data underlying **Figure 2E** we compared the distribution of the maximum generation reached by each family. For these procedures, it sufficed to follow the same rationale as for the tests related to **Figure 2D**, but for the dataset $D = (\tau_i, c_i, s(c_i))_{i=1}^{N}$ of $N$ families, where $\tau_i$ is the maximum generation of the $i$ th family. In particular, the testing statistic $T$ was defined as in **Equation 2**, and the subsequent p-value was estimated as in **Equation 3**.

To challenge the null hypotheses that differentiation without division was independent of the culture condition for the data underlying **Figure 2F**, we compared the proportions of cell types of undivided cells (i.e., those in generation 0). For these procedures, it sufficed to follow the same rationale as for the tests related to **Figure 2D, E**, with $D = (\tau_i, c_i, s(c_i))_{i=1}^{N}$ the sequence of $N$ families in generation 0, where $\tau_i$ identifies the type of the unique cell in family $c_i$. In particular, the testing statistic $T$ was defined as in **Equation 2**, and the subsequent p-value was estimated as in **Equation 3**.

To challenge the null hypotheses that the pattern of first division was independent of the culture condition, for the data underlying **Figure 2G** we compared the proportion of division types among families recovered with two cells in generation 1. For these procedures, it sufficed to follow the same rationale as for the tests in **Figure 2D–F**, with $D = (\tau_i, c_i, s(c_i))_{i=1}^{N}$ as the dataset of $N$ families with two cells generation 1, where $\tau_i$ records the pattern of division of the family $c_i$ as one out of four possibilities (outlined in **Figure 2G**). The test statistic $T$ was defined as in **Equation 2**, and the subsequent p-value was estimated as in **Equation 3**.

For the data in a given time point (24 or 48 hr) underlying **Figure 4B**, we investigated the family effect on differentiation by challenging the null hypotheses that differentiation diversity among cells from the same ancestor type was independent of familial membership. In particular, as the cells from the data were found in different generations, we sought to take into account that division may have had an impact on differentiation (**Figure 3D**). These data were identified by the sequence $D = (\tau_i, g_i, c_i)_{i=1}^{N}$ of the $N$ cells from the same progenitor, with $\tau_i, g_i, c_i$ recording the type, the generation, and the family label, respectively, of the $i$ th cell. To test the null hypothesis by permutation, the set of invariant transformations $Q$ for $D$ should permute, across families, only cells that were found in the same generation. To this end, $Q$ was generated by the functions $\pi_g$ for $g \in G = \{g_i, i = 1, \ldots, N\}$, such that

$$\pi_g(i) = \begin{cases} \tilde{\pi}_g(i) \; if \; g_i = g \\ i \quad otherwise \end{cases}, \tag{4}$$

where $\tilde{\pi}_g$ is any permutation of the set $\{i = 1, \ldots, N : g_i = g\}$. Then $D^{\pi} = (\tau_i, g_i, c_{\pi(i)})_{i=1}^{N}$. To measure family differentiation diversity, we defined the statistic $T$ for the average number of cell types per family, that is,

$$T(D) = \frac{\sum_{c=1}^{M} \sum_{j \in J} \chi(j \in \mathfrak{T}_c)}{M}, \tag{5}$$

where $\{1, \ldots, M\}$ is the set of all family labels, $J = \{\tau_i : i = 1, \ldots, N\}$ is the set of all cell types observed, and $\mathfrak{T}_c = \{\tau_i, i = 1, \ldots, N : c_i = c\}$. In this case, the alternative hypothesis posited that familial relationship induced a more homogeneous differentiation in terms of cell types, leading to a decreased

number of cell types expected per families $T(D)$. For this reason, by Monte Carlo approximation, we estimated the p-value for the left-tailed test as

$$\hat{p}_B^l = \frac{1 + \sum_{i=1}^{B} \chi(T(D) \geq T(D^{\pi_i}))}{1 + B},$$ (6)

where $\pi_1, \ldots, \pi_B$ are sampled uniform and independent sampled elements from $Q$.

Using the data underlying *Figure 4D*, we wished to challenge the null hypotheses that family progression is independent of ancestral expression levels (CD48, c-Kit, Sca-1). For notational purposes, the data were represented as a sequence $D = (\tau_i, g_i)_{i=1}^{N}$ of $N$ families where the $i$ th family was identified by expression level $\tau_i$, relative to one marker; maximum generation of its offspring $g_i$. Given the set of maximum generations attained, $J = \{g_i, i = 1, \ldots, N\}$, we partitioned the data into $D_j = (\tau_i, i = 1 \ldots, N : g_i = j) = (\tau_{ji})_{i=1}^{n_j}$ collections of size $n_j$, for $j \in J$. Thus, we sought to test the null hypothesis that the variables $\tau_{ji}$ are identically distributed, against the alternative hypothesis that, given $m_j$ the median of the distribution from which the elements of $D_j$ are drawn, for every $k, h \in J$ such that $k \leq h$, are either increasing

$$m_k \leq m_h$$

or decreasing

$$m_k \geq m_h,$$

where at least one inequality must be strict. To this end, the statistic $T$ of the data $D$ was defined from the Jonckheere's trend test, that is,

$$T(D) = \sum_{j<l j, l \in J} \sum_{a=1}^{n_j} \sum_{b=1}^{n_l} \chi(\tau_{lb} > \tau_{ja}).$$ (7)

Under the null hypothesis that the variables $\tau_{ji}$ are identically distributed, $D$ is equally likely as a dataset $D^\pi = (\tau_i, g_{\pi(i)})_{i=1}^{N}$ transformed by the action of any permutation $\pi \in Q$ of the set of family labels $\{1, \ldots, N\}$. As a consequence, using Monte Carlo approximation we estimated the p-value for the two-tailed test as

$$\hat{p}_B^t = 2 \min \left( 0.5, \frac{1 + \min(\sum_{i=1}^{B} \chi(T(D) \leq T(D^{\pi i})), \sum_{i=1}^{B} \chi(T(D) \geq T(D^{\pi i})))}{1 + B} \right)$$ (8)

where $B = 250,000$ and $\pi_1, \ldots, \pi_B$ were uniformly and independently sampled from $Q$.

For the data underlying *Figure 4E*, we sought to challenge the null hypotheses that family differentiation to a certain cell type (SLAM-HSC, PreMegE, GMP, CMP/MEP, MPP) is independent of ancestral expression levels (CD48, c-Kit, Flt3, Sca-1). For these procedures, it sufficed to follow the same rationale as for the tests used for the data in *Figure 4D*, but for the dataset $D = (\tau_i, g_i)_{i=1}^{N}$ of $N$ families, where, for the $i$ th family, $\tau_i$ is the expression level from one marker of its ancestor cell, while $g_i = 1$ if its offspring was detected having at least one cell of the cell type under consideration, $g_i = 0$ otherwise. In particular, the testing statistic $T$ was defined as in *Equation 7*, and the subsequent p-value was estimated as in *Equation 8*.

When multiple hypotheses were tested from the same data, the family-wise error rate was controlled using Holm–Bonferroni method (*Lehmann and Romano, 2006*). As such, given the ordered p-values from $k$ simultaneous tests $p_{B1}^t \leq \ldots \leq p_{Bk}^t$, the $i$ th p-value was adjusted and recalculated as

$$\min\left((k + 1 - i)\hat{p}_{Bi}^t, 1\right).$$

## Beta-binomial model for family concordance

To quantify the correlation in decisions of cells to continue to divide or cease dividing, for cells from the same generation and descending from the same ancestor, we employed a stochastic mathematical model that was first described in.

In biological terms, in this mathematical model, cells divide or stop dividing with a certain probability that is correlated between cells from the same family and generation. When sampling, cells that divide less tend to be detected less, simply because cells that divide more are more abundant. Thus, it is important to take sampling into account when measuring the range of division within a family. In this model, cells have a certain probability to be measured defined by the recovery rate. The correlation coefficient that links the division of cells within the same family is fitted to the data and is used to evaluate the degree of correlation in division within families, the so-called concordance in division.

In more detail, the model is directly parameterized by the data, apart from one variable that encapsulates the correlation in decision-making that is fit to the data. In particular, with $n$ being the maximum generation recorded, let $p_i \in [0, 1]$ for $i = 0, \dots, n$ denote the proportion of cells that divide from generation $i$ to the next, which is determined from the data as follows. Set $z_i$ the total number of cells recovered in generation $i$, the $p_i$ were estimated by

$$\hat{p}_i = \frac{\sum_{j=i+1}^n z_j 2^{-j}}{\sum_{j=i}^n z_j 2^{-j}} \tag{9}$$

for $i = 0, \dots, n-1$ and by $p_n = 0$.

In this model, given $k_i$ the number of cells from a particular family that reach generation $i$, the number of cells that continue on to divide to $i+1$ follow a beta-binomial distribution with parameters $k_i$, $a_i = p_i(1-\rho)/\rho$, and $b_i = (1-p_i)(1-\rho)/\rho$, namely $\beta(k_i, a_i, b_i)$, where $\rho \in [0, 1]$ is a free parameter. In particular, each family is generated recursively by setting $k_0 = 1$ and defining $k_{i+1} = 2\beta(k_i, a_i, b_i)$. As in the experimental system, not all cells are recovered, but the proportion that can be determined either by beads or well-volume recovered, on generating a family with the model, we accounted for sampling effect by sub-sampling each cell from a family with probability $r = 0.71$ independently of all other cells. The beta-binomial model interpolates between cells deciding to divide again independently of one another if $\rho = 0$, and when they are perfectly aligned, all making the same division decision, which occurs when $\rho = 1$. A value between 0 and 1 reflects the level of concordance within each family in division-progression decision-making, but, by construction, irrespective of the values of $p_i$, that determines the population distribution among the generations. Defining the family range as the difference between maximum and minimum generations in which the cells from a family are recovered, the best-fit $\rho$ was determined to be the value that maximized the likelihood of recapitulating the family range distribution in the data.

## Acknowledgements

We would like to thank the members of Institut Curies Flow Facility for their help with setting up the flow cytometry experiments, the members of the Institut Curie Animalerie for their care for our experimental animals, Jean-Luc Villeval for counting the methylcellulose colonies, Prof. Phil Hodgkin (Walter and Eliza Hall Institute of Medical Research) and Dr Julia Marchingo (University of Dundee) for their advice on setting up the experiment, and Stefania Pan and Emiliano Lancini (Université Paris 13) for their advice on optimization problems in graph theory. The present study was supported by an ATIP-Avenir grant from CNRS and Bettencourt-Schueller Foundation (to LP) and two grants from the *Labex CelTisPhyBio* (ANR-10-LBX-0038) and Idex Paris-Science-Lettres Program (ANR-10-IDEX-0001-02 PSL) (to LP).

## Additional information

### Funding

| Funder | Grant reference number | Author |
|---|---|---|
| Fondation Bettencourt Schuel- | ATIP-Avenir | Leïla Perié |

ler

| Centre National de la Re-cherche Scientifique | ATIP-Avenir | Leïla Perié |
| --- | --- | --- |
| Labex Cell(n)Scale | ANR-10-LBX-0038 | Leïla Perié |
| Idex Paris Sciences et Lettres | ANR-10-IDEX-0001-02 PSL | Leïla Perié |

The funders had no role in study design, data collection and interpretation, or the decision to submit the work for publication.

## Author contributions

Tamar Tak, Conceptualization, Data curation, Formal analysis, Validation, Investigation, Visualization, Methodology, Writing - original draft, Writing - review and editing; Giulio Prevedello, Conceptualization, Software, Formal analysis, Investigation, Visualization, Writing - review and editing; Gaël Simon, Formal analysis, Investigation, Methodology; Noémie Paillon, Investigation, Methodology; Camélia Benlabiod, Caroline Marty, Isabelle Plo, Investigation; Ken R Duffy, Conceptualization, Formal analysis, Supervision, Investigation, Writing - original draft, Writing - review and editing; Leïla Perié, Conceptualization, Supervision, Investigation, Funding acquisition, Project administration, Writing - original draft, Writing - review and editing

## Author ORCIDs

Tamar Tak ⬤ https://orcid.org/0000-0002-5959-7927
Giulio Prevedello ⬤ https://orcid.org/0000-0002-9857-2351
Noémie Paillon ⬤ http://orcid.org/0000-0002-6848-3016
Ken R Duffy ⬤ https://orcid.org/0000-0001-5587-9356
Leïla Perié ⬤ https://orcid.org/0000-0003-0798-4498

## Ethics

Animal experimentation: All the experimental procedures were approved by the local ethics committee (Comité d'Ethique en expérimentation animale de l'Institut Curie) under approval number DAP 2016 006.

## Decision letter and Author response

Decision letter https://doi.org/10.7554/eLife.60624.sa1
Author response https://doi.org/10.7554/eLife.60624.sa2

# Additional files

## Supplementary files

• Source data 1. Complete dataset used in the article for each analyzed cell, its generation, assigned cell type, family membership, marker fluorescence intensities at the time point of analysis, and marker fluorescence intensities of its ancestor cell during sort.

• Source code 1. All code used for generation of figures and statistical testing.

• Transparent reporting form

## Data availability

All data generated or analysed during this study are included in the manuscript and supporting files. Source data has been provided for Figures 1-4.

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
