## [Decision Letter]

Acceptance summary:

Tak et al. address an important problem in stem cell biology, which is how stem cells (in this case, hematopoietic stem cells, or HSCs), most of which are known to be biased or restricted in their lineage output, acquire their lineage bias. Specifically, the authors sought to uncouple the process of division and differentiation from divergent instructive signals from a niche by culturing single cells (or actually groups of 4 cells) ex vivo under defined conditions (i.e. the same extrinsic instructive signals) and examining cell division and associated lineage decisions. The authors make several conclusions from their studies. One important conclusion is that daughters of a single ancestor HSC or progenitor and their progeny divided at very similar rates, which differed between ancestral cell families, suggesting that the division rate is an intrinsic property of the ancestor cell that was passed along to its daughters. The authors also tried to link the properties of daughter cells to the levels of cell surface markers on the ancestral clone using the original index sorting information, but found that cell surface marker levels only partially correlated with the division and differentiation properties.

Decision letter after peer review:

Thank you for sending your article entitled "HSPCs display within-family homogeneity in differentiation and proliferation despite population heterogeneity" for peer review at *eLife*. Your article is being evaluated by 2 peer reviewers, and the evaluation has been overseen by a Reviewing Editor and Utpal Banerjee as the Senior Editor.

*Reviewer #1:*

This study reports an in vitro cell labeling method combined with mathematical modeling to simultaneously track the number of cell divisions and the differentiation progress of single murine hematopoietic stem cells and progenitors. The authors conclude that cells derived from a common ancestor have statistically significant concordant proliferation and differentiation.

I found the paper was difficult to understand in parts, not just because the mathematical modeling is outside my expertise.

1. Although the immunophenotypes used to isolate the populations from fresh bone marrow are well established, reliance on immunophenotype in cultured cells to define functional status is problematic.

2. The biology of HSC and progenitor cells is dealt with superficially-an example in point is the casual statement that the immunophenotypic data show "dedifferentiation" of progenitors to HSC, a phenomenon that is not accepted by the field; proof of this heresy would require use of more rigorous assays than those used here.

3. The assignment of immunophenotype based on re-staining cells 24-48 hours after original staining requires careful controls to ensure antibody associated fluorescence is not a carryover from prior antibody staining.

4. As the culture conditions (supraphysiologic concentrations of cytokines in cell suspension without ECM or stroma) do not reflect the complexity of the normal microenvironment the relevance of the findings to normal HSPC biology is limited.

*Reviewer #2:*

Tak et al. address an important problem in stem cell biology, which is how stem cells (in this case hematopoietic stem cells, or HSCs), most of which are known to be biased or restricted in their lineage output, acquire their lineage bias. Specifically, the authors sought to uncouple the process of division and differentiation from divergent instructive signals from a niche by culturing single cells (or actually groups of 4 cells) ex vivo under defined conditions (i.e. the same extrinsic instructive signals) and examining cell division and associated lineage decisions. The authors make several conclusions from their studies. One important conclusion is that daughters of a single ancestor HSC or progenitor and their progeny divided at very similar rates, which differed between ancestral cell families, suggesting that the division rate was an intrinsic property of the ancestor cell that was passed along to its daughters. The authors also tried to link the properties of daughter cells to the levels of cell surface markers on the ancestral clone using the original index sorting information, but found that cell surface marker levels only partially correlated with the division and differentiation properties.

The study required sophisticated statistical analyses. An issue is that the authors defined progenitors that differentiated from the original ancestral cell using cell surface markers that were originally defined using freshly isolated cells coupled with functional assays. The authors use these markers to define progenitor types following one or two days of ex vivo culture. Markers can change in culture, and the authors did not demonstrate with functional assays that the markers reliably identified progenitors that have been in culture for 1-2 days. This does not impact their conclusions about the division history, but it may impact the proper identification of the more differentiated progeny of the ancestral cell.

Has the faithfulness of the marker expression in cultured cells been verified by other laboratories? If so, please provide a reference. If not, the authors should explain how they determined that marker expression correctly identified different progenitor types in culture.

---

## [Author Response]

Reviewer #1:This study reports an in vitro cell labeling method combined with mathematical modeling to simultaneously track the number of cell divisions and the differentiation progress of single murine hematopoietic stem cells and progenitors. The authors conclude that cells derived from a common ancestor have statistically significant concordant proliferation and differentiation.I found the paper was difficult to understand in parts, not just because the mathematical modeling is outside my expertise.1. Although the immunophenotypes used to isolate the populations from fresh bone marrow are well established, reliance on immunophenotype in cultured cells to define functional status is problematic.

We thank the Reviewer for raising this important point. Most of our phenotypic definitions were inspired by Pronk et al., Cell Stem Cell 2017. The comment from reviewer 2 made us realize that our naming was not fully transparent (in particular for Slam+MEP) so we have changed the naming of the progenitors phenotypically defined after culture.

We agree that the cell surface markers we used are most often used for identification of cell populations in freshly isolated bone marrow rather than cultured cells (as it is the case in Pronk et al). We have now performed semi-solid cultures (methylcellulose and CFU-MK assays) to functionally assess the differentiation capacity of the different cell populations defined with our phenotypic definitions after 48h of culture (SCF, TPO, IL3 and IL6). As now described in the manuscript page 19 and shown in the new panels B and C for figure 2, these assays have shown that the differentiation capacity of these progenitors after culture is very close to the fresh ones.

2. The biology of HSC and progenitor cells is dealt with superficially-an example in point is the casual statement that the immunophenotypic data show "dedifferentiation" of progenitors to HSC, a phenomenon that is not accepted by the field; proof of this heresy would require use of more rigorous assays than those used here.

While we can see how the reviewer may have come to that conclusion, we respectfully disagree that our statement was casual as its inclusion was actually entirely considered. However, the finding about phenotypic dedifferentiation is not the focus of our study. Carrying in vivo transplantation involved work that goes beyond its remit and, therefore, we have entirely removed the comment about dedifferentiation in our manuscript.

3. The assignment of immunophenotype based on re-staining cells 24-48 hours after original staining requires careful controls to ensure antibody associated fluorescence is not a carryover from prior antibody staining.

We were also concerned about this issue and so had already checked that the fluorescence is lost within 24h of culture. We have added this data as Figure 2—figure supplement 2.

4. As the culture conditions (supraphysiologic concentrations of cytokines in cell suspension without ECM or stroma) do not reflect the complexity of the normal microenvironment the relevance of the findings to normal HSPC biology is limited.

We agree that our study in vitro does not reproduce the in vivo complexity. in vitro studies are, of course, never physiological but are employed as reasonable models in which to study effects that are not assessable in vivo. Our approach to ameliorating those limitations was to use two distinct culture conditions, which demonstrate that the observations on concordance in division and differentiation are consistent in both. We disagree with the reviewer, and believe these results are still significant for HSPC biology.

Reviewer #2:Tak et al. address an important problem in stem cell biology, which is how stem cells (in this case hematopoietic stem cells, or HSCs), most of which are known to be biased or restricted in their lineage output, acquire their lineage bias. Specifically, the authors sought to uncouple the process of division and differentiation from divergent instructive signals from a niche by culturing single cells (or actually groups of 4 cells) ex vivo under defined conditions (i.e. the same extrinsic instructive signals) and examining cell division and associated lineage decisions. The authors make several conclusions from their studies. One important conclusion is that daughters of a single ancestor HSC or progenitor and their progeny divided at very similar rates, which differed between ancestral cell families, suggesting that the division rate was an intrinsic property of the ancestor cell that was passed along to its daughters. The authors also tried to link the properties of daughter cells to the levels of cell surface markers on the ancestral clone using the original index sorting information, but found that cell surface marker levels only partially correlated with the division and differentiation properties.The study required sophisticated statistical analyses. An issue is that the authors defined progenitors that differentiated from the original ancestral cell using cell surface markers that were originally defined using freshly isolated cells coupled with functional assays. The authors use these markers to define progenitor types following one or two days of ex vivo culture. Markers can change in culture, and the authors did not demonstrate with functional assays that the markers reliably identified progenitors that have been in culture for 1-2 days. This does not impact their conclusions about the division history, but it may impact the proper identification of the more differentiated progeny of the ancestral cell.Has the faithfulness of the marker expression in cultured cells been verified by other laboratories? If so, please provide a reference. If not, the authors should explain how they determined that marker expression correctly identified different progenitor types in culture.

We thank the reviewer for raising this important point. First, the Reviewer’s comment made us realize that we were not using the same names than the one used in the literature (eg Pronk et al., 2007; Grinengo et al., 2018) which was misleading. We have changed the nomenclature to follow a similar scheme to Pronk et al., 2007 (e.g. Slam+MEP is now PreMegE). We agree that the cell surface markers we used are most often used for identification of cell populations in freshly isolated bone marrow than cultured cells (as it is the case in Pronk et al).

We have now performed semi-solid cultures (methylcellulose and CFU-MK assays) to functionally assess the differentiation capacity of the different cell populations defined with our phenotypic definitions after 48h of culture (SCF, TPO, IL3 and IL6). As now described in the manuscript page 19 and included in figure 2,these assays have shown that the differentiation capacity of these progenitors after culture is very similar to the fresh ones.